# A Method of Accelerating the Convergence of Computational Fluid Dynamics for Micro-Siting Wind Mapping

**Hyun-Goo Kim** 

New and Renewable Energy Resources & Policy Center, Korea Institute of Energy Research, Daejeon 34129, Korea; hyungoo@kier.re.kr; Tel.: +82-42-860-3376

**Abstract:** To assess wind resources, a number of simulations should be performed by wind direction, wind speed, and atmospheric stability bins to conduct micro-siting using computational fluid dynamics (CFD). This study proposes a method of accelerating CFD convergence by generating initial conditions that are closer to the converged solution. In addition, the study proposes the 'mirrored initial condition' (IC) using the symmetry of wind direction and geography, the 'composed IC' using the vector composition principle, and the 'shifted IC' which assumes that the wind speed vectors are similar in conditions characterized by minute differences in wind direction as the well-posed initial conditions. They provided a significantly closer approximation to the converged flow field than did the conventional initial condition, which simply assumed a homogenous atmospheric boundary layer over the entire simulation domain. The results of this study show that the computation time taken for micro-siting can be shortened by around 35% when conducting CFD with 16 wind direction sectors by mixing the conventional and the proposed ICs properly.

**Keywords:** computational fluid dynamics; micro-siting; wind mapping; initial condition; convergence

## 1. Introduction

In a wind resource assessment (WRA) conducted to construct a wind farm, at least one or more meteorological towers should be installed at representative meteorological points within the candidate area, and the weather conditions, including seasonal changes, should be measured for at least one year. The measurement data should be converted to statistical data on wind resources at the hub height of the wind turbine at the installation points to be able to calculate the amount of wind power generated. However, the wind field is not homogeneous and varies according to the terrain. Thus, the meteorologically valid range of data measured with the meteorological tower is limited near the measurement point. Accordingly, wind flow modeling should be performed to identify the wind resources across a wide range of candidate areas for the construction of a wind farm. In other words, the wind speed distribution by wind direction should be mapped in the candidate area [1].

Wind flow modeling refers to a method of analyzing the Navier–Stokes equation, which is the governing equation of atmospheric wind flow. The modeling methods are divided into a linear method that is suitable for flat terrain, and a computational fluid dynamics (CFD) method that is suitable for complex terrain. As regards the representative software for each method, WAsP and WindSim can be used. Other methods, such as the interpolation method, continuity equation analysis method, and meso-scale numerical weather prediction (NWP) method, are also available, but the CFD method is now becoming the mainstream method [2,3].

The wind turbine layout that most effectively minimizes wake losses while maximizing energy production should be determined when the wind turbines are laid out in the candidate area. Then, the wind farm's annual energy production (AEP, MWh) can be calculated using the following equation [4,5]:

$$\text{AEP} = 8760 \times \sum_{i=1}^{nD} \sum_{j=1}^{nU} \sum_{k=1}^{nL} \sum_{l=1}^{nT} f_{ijkl} \cdot P_{ijkl} \tag{1}$$

In the above equation, $nD$, $nU$, $nL$, and $nT$ refer to the number of wind direction sectors, the number of wind speed bins, the number of atmospheric stability intervals, and the number of wind turbines, respectively. For example, the wind speed range is divided into a series of intervals known as bins.

Here, $f_{ijkl}$ refers to the probability density function that exhibits the frequency of occurrence of the $j$-th wind speed bin in the $i$-th wind direction sector and the $k$-th atmospheric stability interval in the $l$-th wind turbine location, while $P_{ijkl}$ refers to the power output (kW). To calculate the AEP from Equation (1) after determining the wind turbine layout in the wind farm area, $f_{ijkl}$ and $P_{ijkl}$ should be identified first.

$P_{ijkl}$ is given by the power curve of the wind turbine, and $f_{ijkl}$ can be expressed by the Weibull distribution function, which is a probability distribution function of wind speed. $f_{ijkm}$ can be evaluated by long-term correction after the measurement data have been extrapolated up to the wind turbine hub height at the installation location ($l = m$) of the meteorological tower. Either way, the results of a simulation of a mesoscale numerical weather prediction can be downscaled [6]. However, to determine $f_{ijkl}$ at an arbitrary location within the wind farm area, the meteorological correlation between $f_{ijkm}$ and $f_{ijkl}$ should be evaluated by wind flow modeling, which process is called micro-siting.

$f_{ijkl}$ at the arbitrary location can be calculated (as presented in the following equation) by applying the acceleration or deceleration ratio of the wind speed $V$ predicted in the numerical modeling and $f_{ijkm}$ at the meteorological tower's location.

$$f_{ijkl} = \frac{V_{ijkl}}{V_{ijkm}} f_{ijkm} \tag{2}$$

If a three-dimensional wind flow field that is changing in time should be predicted by transient simulation, an hourly numerical analysis of 8760 h is needed for one year, which requires a large amount of computation load and time. However, rather than conducting a numerical analysis for all 8760 h, if a typical case for each interval of wind direction, wind speed and atmospheric stability is analyzed by dynamical downscaling while assuming that the wind field at each hour is independent, and if the frequency of occurrence for each case is summed after multiplying it by the weighting factor, then the calculation load and time can be reduced significantly [7]. Thus, in such a case, wind flow modeling is performed with all of the wind condition sectors that can occur meteorologically and statistically, i.e., $nD \times nU \times nL$ cases, to reduce the calculation load and time.

If dynamical downscaling is set to 16 wind direction sectors, five wind speed bins, and five atmospheric stability intervals, a total of $16 \times 5 \times 5 = 400$ cases are to be simulated, which equals about 4.6% of the 8760 hourly cases. Nonetheless, 400 cases will still require a considerable calculation load and time for CFD. Here, $nD = 16$ means 16 wind direction sectors at 22.5° intervals, and $nU = 5$ means five wind speed bins from 0 m/s to 25 m/s, which is the cut-out wind speed that stops the operation of the wind turbine, at 5 m/s intervals. $nL = 5$ means that the Monin–Obukhov length is divided by five intervals from $-\infty$ to $+\infty$.

The CFD simulation of the atmospheric flow field means a process of iterative numerical analysis of the algebraic equations that satisfy the governing equation, i.e., the Reynolds-Averaged Navier–Stokes (RANS) equation, by all the flow field variables defined in each of the finite volume cells constituting the flow domain. To solve the partial differential RANS equation, boundary conditions should be imposed at the external surfaces of a three-dimensional physical space together with the initial conditions inside a three-dimensional physical space.

Since the RANS equation is an elliptic partial differential equation, there exists a unique solution regardless of the initial conditions in the case of a steady-state problem. However, an ill-posed initial condition would cause a divergence or deceleration of convergence in a numerical analysis [8]. Conversely, an acceleration of convergence can be expected when a good approximation of the initial conditions is imposed [9]. In particular, micro-siting wind mapping requires a considerable number of cases, such as the 400 described above. Thus, the assumption of the initial conditions can influence the overall computation time.

For an atmospheric flow field, a method that assumes the atmospheric boundary layer profiles of wind speed and turbulence, is slightly more effective in shortening the convergence speed than one that assumes a uniform flow field and turbulence field [10]. Either a potential or Euler flow field whose computation time is relatively shorter than the CFD is analyzed first to assume an accurate initial flow field, and its convergence solution may be used as the initial condition of the CFD [11,12].

This study proposes a novel method of accelerating the convergence speed in the CFD by assuming initial conditions that are closer to the real flow field, i.e., a solution that is converged as much as possible using the geometric characteristics in the atmospheric flow field according to the wind direction when producing the micro-siting wind map [13], and which proves the quantitative effect through real application cases.

## 2. Methods and Data

### 2.1. Method of Generating the Initial Conditions

To perform micro-siting wind mapping for a wind resource assessment, CFD simulations of a total number of $nD \times nU \times nL$ cases should be conducted with regard to the $nD$ wind direction sectors, $nU$ wind speed bins, and $nL$ atmospheric stability intervals. In the wind energy industry, the Reynolds number invariance can be assumed in the case of a turbulent flow of a high Reynolds number without accompanying a flow separation [14,15]. Therefore, the number of CFD simulation cases can be reduced significantly using the $nU = 1$ assumption. If the neutral atmospheric stability is additionally assumed, then it satisfies $nL = 1$; thus the CFD can be performed with only the $nD$ wind direction sectors [16]. Figure 1 shows the procedure for micro-siting wind mapping via a CFD simulation of the $nD$ wind direction sectors.

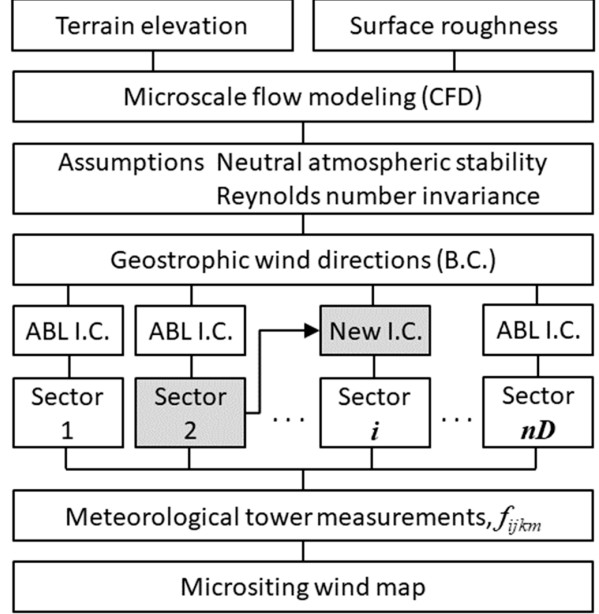

**Figure 1.** Flow chart of the micro-siting wind mapping procedure.

This study proposes a method of accelerating the convergence speed by assuming initial conditions that are very close to the converged solution, i.e., the real flow field, using the geometric characteristics of the atmospheric flow field and the topography when the CFD is conducted with regard to the *nD* wind direction sectors, as follows:

First, the geometric symmetry of the atmospheric flow field and the topography can be employed. For example, a northerly wind exhibits geometric symmetry with a southerly wind that has a 180° angular difference along the west–east axis. The initial condition that is reconfigured using this property is named the 'mirrored initial condition (IC)'.

This characteristic is expressed by the following equations:

$$\overrightarrow{V}(\theta) \cong -\overrightarrow{V}(\theta \pm 180) \tag{3}$$

$$\nabla P(\theta) \cong -\nabla P(\theta \pm 180) \tag{4}$$

where $\overrightarrow{V}(\theta)$ is the wind vector of the wind directional sector $\theta$ case, and $\nabla P$ is the pressure gradient in the wind vector direction.

Second, the wind vector of the wind direction sector $\theta$ case can be calculated mathematically by the vector composition of the wind vectors $\theta + d\theta$ and $\theta - d\theta$ when $d\theta$ is small enough. The initial condition that is reconfigured using this property is called the 'composed IC'.

This characteristic is expressed by the following equations:

$$\overrightarrow{V}(\theta) \cong \frac{\left|\overrightarrow{V}(\theta - d\theta)\right| + \left|\overrightarrow{V}(\theta + d\theta)\right|}{2} \cdot \frac{\overrightarrow{V}(\theta - d\theta) + \overrightarrow{V}(\theta + d\theta)}{\left|\overrightarrow{V}(\theta - d\theta) + \overrightarrow{V}(\theta + d\theta)\right|} \tag{5}$$

$$\nabla P(\theta) \cong \frac{1}{2}\{\nabla P(\theta - d\theta) + \nabla P(\theta + d\theta)\} \tag{6}$$

Third, the wind vector of the wind direction sector $\theta$ case would be similar to the wind vector of the wind direction sector $\theta \pm d\theta$ case, which is rotated by as much as $d\theta$ with respect to the wind direction $\theta$. The initial condition that is reconfigured using this property is called the 'shifted IC'.

This characteristic is expressed by the following equations:

$$\overrightarrow{V}(\theta + d\theta) \cong \left|\overrightarrow{V}(\theta)\right| \cdot \overrightarrow{i}(\theta + d\theta) \tag{7}$$

$$\nabla P(\theta + d\theta) \cong \nabla P(\theta) \tag{8}$$

In the above equation, $\overrightarrow{i}$ refers to the unit vector. That is, $\left|\overrightarrow{i}\right| = 1$.

### 2.2. Wind Mapping Steps for Convergence Acceleration

According to the international standard for wind energy, IEC 61400-1, the interval of the wind direction sectors shall be 30° or less [17]. Therefore, most wind resource assessments consider either 12 or 16 wind direction sectors. In case of $nD = 16$, convergence can be accelerated by conducting a CFD simulation using the following steps, by employing new initial conditions in Section 2.1 and Figure 1.

Step 1: CFD simulation is performed by applying a conventional initial condition for the four wind direction sectors, i.e., N, NE, E, and SE, thereby, obtaining the converged solution.

Step 2: The mirrored IC is applied with regard to the four wind direction sectors, i.e., S, SW, W, and SW, which have 180° symmetry with the above wind directions respectively, to perform the CFD simulation and thereby, obtain the convergence solution.

Step 3: Either the composed IC or the shifted IC is applied with regard to the other eight wind directions, i.e., NNE, ENE, ESE, SSE, SSW, SWS, NWN, and NNW, to conduct the CFD simulation and thereby, obtain the convergence solution.

As described above, the generated initial conditions can be applied to accelerate convergence for the 16 wind directions, except for the first four wind direction sectors out of the 16 wind directions.

### 2.3. Verification of the Convergence Acceleration Effect

To verify the effectiveness of this method of generating the initial conditions for convergence acceleration, WindSim, a representative CFD-based micro-siting software, was used. WindSim solves the RANS equations with the finite volume method and uses a meshed grid system of terrain [18]. A computer equipped with a Xeon CPU X5460 3.16 GHz with 32 GB RAM was used for testing, and the Hundhammerfjellet wind farm in Norway (which was included as a basic example in "WindSim: Getting Started") was used as the verification case. The grid system configured for the CFD simulation in the Hundhammerfjellet region was as follows: domain size = 15 km × 15 km, fine grid of 75 m × 75 m (200 × 200 × 25 = 1 million cells). For reference, this setting satisfies grid independency and is suitable for application to a wind farm design in terms of spatial resolution.

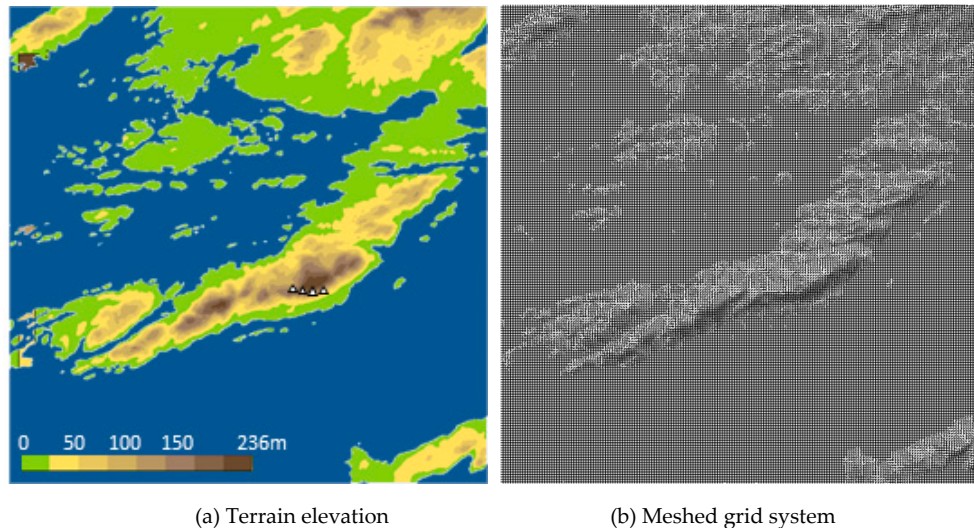

(a) Terrain elevation　　　　　　　　　　　　(b) Meshed grid system

**Figure 2.** The computational domain around the Hundhammerfjellet wind farm (triangles: wind turbines, cobalt blue region: sea).

Since the atmospheric flow has a very large topographic scale generally, it has a high Reynolds number, and since the ground surface is rough, Reynolds number invariance can be assumed. To verify this assumption, the geostrophic wind speeds ($V_{geo}$) in the upper top part of the atmospheric boundary layer were set to 5 m/s, 10 m/s, and 15 m/s. As reference information, the Hundhammerfjellet wind farm is located along the ridge in the central part of the peninsula extending SSW to NNE in the center of the domain, as shown in Figure 2. Assuming that the maximum altitude above sea level (236 meters) of the Hundhammerfjellet ridge is set to a characteristic length, the corresponding Reynolds numbers become $0.8 \times 10^8$, $1.6 \times 10^8$, and $2.4 \times 10^8$, respectively.

## 3. Results and Discussion

### 3.1. Reynolds Number Invariance of the Wind Field

Figure 3a shows the wind speed contour in the northerly wind case, which is normalized with the geostrophic wind speed, meaning that $S = V/V_{geo}$. Furthermore, a low wind speed region was observed below the wind speed ratio $S < 0.2$ on the lee side due to the rapid downslope after a speed-up of more

than $S > 1.0$ at the Hundhammerfjellet ridge in the central part of the computational domain. Figure 3b shows the difference between case $S_{15}$ at a geostrophic wind speed of 15 m/s and case $S_{05}$ at 5 m/s, that is, $dS = S_{15} - S_{05}$. The region which revealed the largest difference in the domain was the deceleration area of wind speed due to the downslope and this effect produced a long transport pattern along the downwind direction.

The minimum value of normalized wind speed difference $dS$ in the deceleration area was –0.09, which meant that wind speed deceleration was predicted more on the ridge's lee side when the Reynolds number was smaller. The relative RMSE of $dS$ in the overall computational domain was just 0.65%. In summary, it was verified that the atmospheric flow field had the characteristic of near invariability according to the Reynolds number overall, although there was a small variation when a rapid wind speed gradient was generated in local regions with a steep topography.

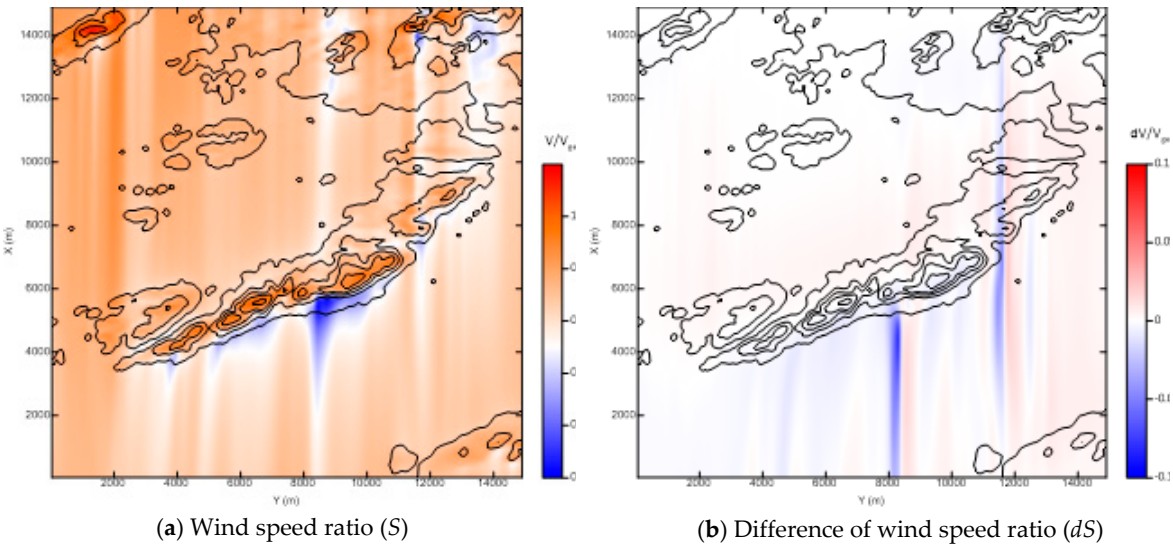

(**a**) Wind speed ratio (*S*)  (**b**) Difference of wind speed ratio (*dS*)

**Figure 3.** Contour plots of wind speed ratio for the northerly wind case (5 m above ground level).

*3.2. Evaluation of New Initial Conditions*

3.2.1. Error Analysis of New Initial Conditions

For the purposes of this study, an error analysis was performed to quantitatively evaluate the method of generating the new initial conditions. An approximation error wind field $d\vec{V}$ is defined as the difference between a converged wind field $\vec{V}_c$, which is assumed to be the true solution, and a guessed wind field $\vec{V}_i$, which is generated by the proposed method, i.e., Equations (3)~(8). If an initial condition is generated using the mirrored IC method, $d\vec{V}$ is as follows:

$$d\vec{V} = \vec{V}_c(\theta) - \vec{V}_i(\theta) = \vec{V}_c(\theta) - \left\{ -\vec{V}_c(\theta \pm 180) \right\} \tag{9}$$

where the subscripts *c* and *i* denote the 'converged solution' and the 'guessed initial condition', respectively.

Figure 4 shows the difference between the converged north-easterly wind field $\vec{V}_c(\text{NE})$ and the guessed wind field $\vec{V}_i(\text{NE}) = -\vec{V}_c(\text{SW})$ using the mirrored IC method with the converged south-westerly wind field $\vec{V}_c(\text{SW})$. The largest error was found in the SW direction of the Hundhammerfjellet ridge, where the terrain slopes steeply, meaning that wind speed acceleration and deceleration occurred for the opposite wind directions, respectively.

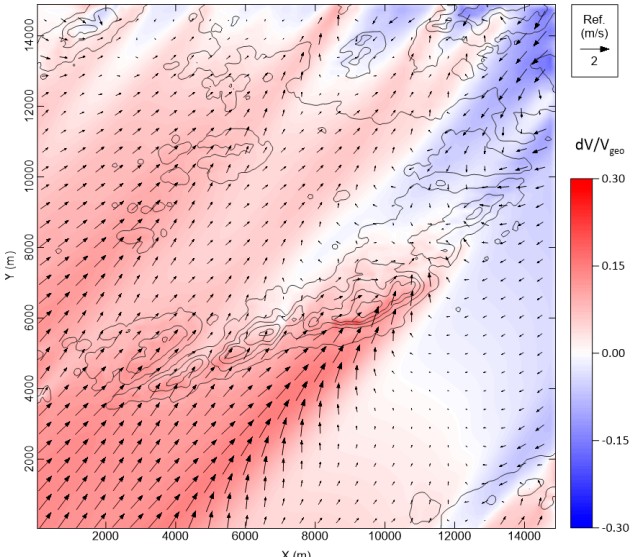

**Figure 4.** Error wind field of the mirrored initial condition for the north-easterly wind case (10 meters above ground level).

Table 1 summarizes the results of the evaluation of the approximation errors between the converged solutions and the guessed initial conditions. In the table, ABL IC stands for the atmospheric boundary layer assumption, which is generally used in micro-siting wind mapping. Note that the results were calculated on the layer at a height of 10 m above ground, where wind field deformation is significant. The proposed initial conditions showed a lower margin of error in all cases than those of the conventional initial condition (ABL) for wind speed and direction. Therefore, the proposed method of generating the initial condition provides a closer approximation to the real wind field than the conventional initial condition. Among them, the composed IC method showed the lowest approximation error.

**Table 1.** Comparison of approximation error of normalized wind speed difference and wind direction difference by the method of generating the initial condition.

| Initial Condition | $dV/V_{geo} \times 100$ (%) | | | $d\theta$ | | |
|:---:|:---:|:---:|:---:|:---:|:---:|:---:|
| | **MBE** | **MAE** | **RMSE** | **MBE** | **MAE** | **RMSE** |
| Mirrored | 3.3 | 6.4 | 8.0 | −0.97 | 2.35 | 3.75 |
| Composed | 4.3 | **4.9** | **5.9** | **0.17** | **2.12** | **3.17** |
| Shifted | **−1.9** | 6.1 | 7.9 | −0.49 | 2.90 | 6.29 |
| ABL | −5.7 | 8.7 | 1.2 | 3.62 | 4.35 | 7.46 |

### 3.2.2. Sensitivity Analysis of Directional Interval

Because a directional interval is involved in the generation of the composed and shifted ICs (Equations (5) and (7)), it is necessary to check the sensitivity of the directional interval to approximation error. As expected, the mean absolute error (MAE) of the approximation errors of wind speed (the dashed lines with hollow circles) and direction (the solid lines with black squares) increased linearly in line with the increase of the directional intervals, as shown in Figure 5. The mean bias error (MBE) and root mean square error (RMSE) showed the same trends.

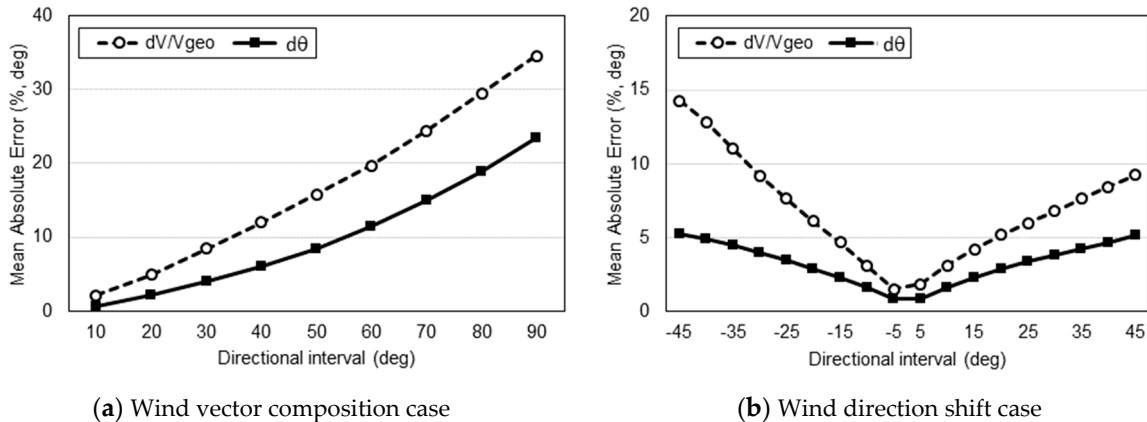

(**a**) Wind vector composition case       (**b**) Wind direction shift case

**Figure 5.** Analysis of the sensitivity of directional intervals to approximation error.

If the maximum error is limited to the MAE of the mirrored IC (Table 1), then the maximum directional intervals for the composed and shifted ICs will be less than 25°, meaning that at least 16 wind directional sectors are needed to apply the new initial conditions.

### 3.3. Convergence Acceleration by the New Initial Conditions

#### 3.3.1. Convergence Acceleration of the Individual CFD Simulation

Figure 6a,b show the improved results of the CFD's convergence speed as the initial conditions are changed from the ABL assumption to the mirrored IC, respectively. Figure 6 shows the history of the field variables at the monitoring point and residuals according to the iteration number. By applying the mirrored IC, the number of iterations that reached convergence was reduced from 100 (ABL IC) to 50 (mirrored IC), meaning an acceleration rate of 50%. In the case of the composed IC or the shifted IC, the acceleration effect of the convergence speed was similar to that of the mirrored IC.

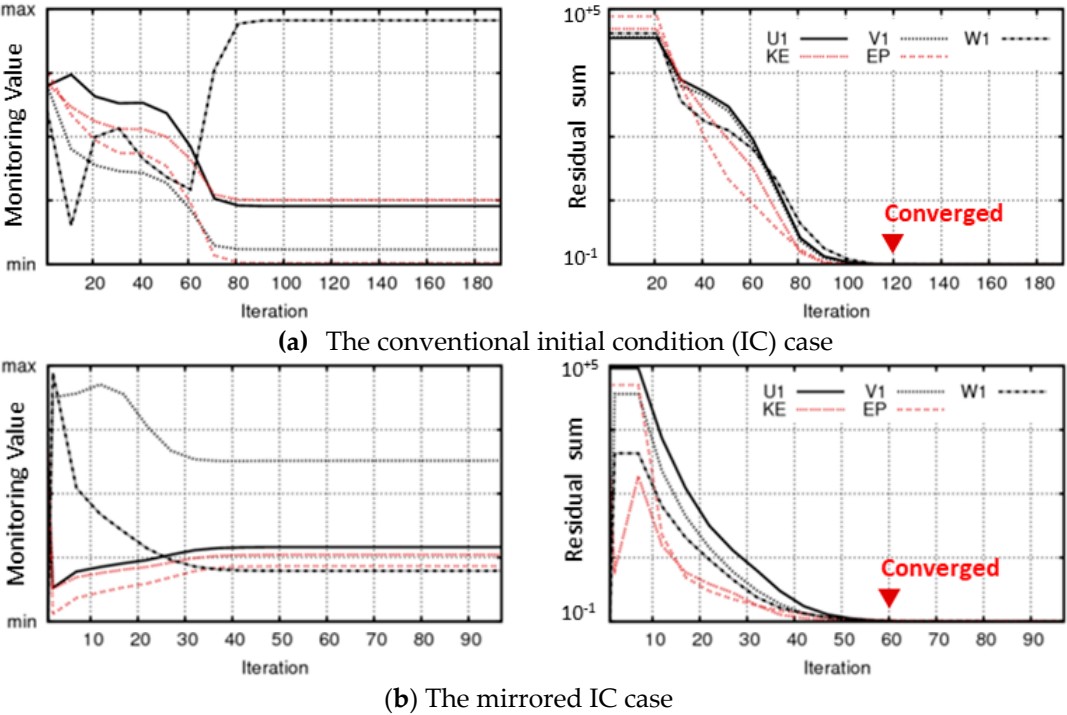

(**a**) The conventional initial condition (IC) case

(**b**) The mirrored IC case

**Figure 6.** Comparison of the convergence history of monitoring values of the field variables (normalized by maximum; left graphs) and the residual values of the field variables (right graphs).

### 3.3.2. Convergence Acceleration of the Overall CFD Simulations

Table 2 shows the quantitative improvements in the convergence speed obtained when the CFD simulation was conducted for the $nD = 16$, $nU = 1$, and $nL = 1$ cases. The overall computation time took 27,150 s when only the ABL was assumed homogeneously in the computation domain. In contrast, the computation time according to the proposed method combining the ABL, mirrored, composed, and shifted ICs (as proposed in Section 2.2) was as follows:

Step 1: For the four wind direction sectors, i.e., N, NE, E, and SE, the ABL IC took 7167 s until convergence.

Step 2: For the four symmetric wind direction sectors, i.e., S, SW, W, and NW, the converged solutions of the previous four sectors of N, NE, E, and SE were used to generate the mirrored IC, and took 3603 s until convergence.

Step 3: For the remaining eight wind direction sectors, i.e., NNE, ENE, ESE, SSE, SSW, SWS, NWN, and NNW, the converged solutions of the previous eight sectors were used to generate either the vector composed IC (e.g., NNE from N+NE) or the shifted IC (e.g., SSW from either S or SSW), and took 6719 s until convergence.

Thus, the overall computation time was 17,489 s, which was 36% shorter than in the conventional case (27,150 s).

**Table 2.** Comparison of the convergence times of the 16 wind direction cases between the conventional initial condition (IC) and the proposed ICs.

| Wind Directional Sector | Convergence Time (s) | | | |
|---|---|---|---|---|
| | Conventional Method | Proposed Method Using New ICs | | |
| | | Step 1 | Step 2 | Step 3 |
| | ABL IC | ABL IC | Mirrored IC | Composed or Shifted IC |
| N | 1740 | 1740 | | |
| NNE | 1837 | | | 919 |
| NE | 1980 | 1980 | | |
| ENE | 1855 | | | 927 |
| E | 1565 | 1565 | | |
| ESE | 1591 | | | 795 |
| SE | 1883 | 1883 | | |
| SSE | 1691 | | | 845 |
| S | 1770 | | 885 | |
| SSW | 1688 | | | 844 |
| SW | 1674 | | 1004 | |
| SWS | 1564 | | | 782 |
| W | 1471 | | 735 | |
| NWN | 1583 | | | 791 |
| NW | 1631 | | 979 | |
| NNW | 1632 | | | 816 |
| Total | 27,150 | 7167 | 3603 | 6719 |
| | | 17,489 | | |

### 3.3.3. Analysis of Sensitivity of the Computation Cell Numbers

In general, the convergence time does not increase linearly according to the number of computation cells. For the purposes of this study, a sensitivity analysis was conducted by increasing the computational cell numbers from half a million up to 4 million to figure out the effect on the convergence time. Figure 7 shows the graph of the empirical equation of the convergence time according to the number of computation cells, in which the convergence time increases as a quadratic function with regard to the number of computation cells (ABL IC was applied). Thus, if 160 min can be reduced when the number of computation cells is 1 million, as presented in Table 2, the shortening time of the convergence speed is estimated to be 50 hours, which is around 20 times of 160 min, if the number of computation cells is increased fourfold to 4 million.

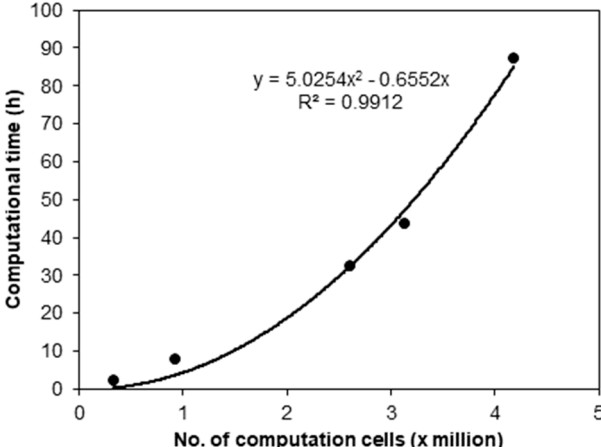

**Figure 7.** Computation time depending on a number of computational cells (for a single sector).

## 4. Conclusions

A large number of CFD simulations need to be performed by wind direction sector, wind speed bin, and atmospheric stability interval to conduct micro-siting in a wind resource assessment. As such, this study proposed a method of shortening the CFD convergence time by assuming initial conditions that are closer to the real flow field. The conclusions obtained in this study are as follows:

(1) Mirrored, composed, and shifted ICs were generated from the converged solutions with regard to the different wind direction sectors by using the geometric similarity or vector composition principle. In the case of the CFD simulations of the 16 wind direction sectors in the Hundhammerfjellet wind farm case, the new ICs showed better convergence performance than that of the conventional ABL IC case, shortening the convergence time by 50%. Compared to the converged solution, the new ICs showed approximation errors of only 4.9%~6.4% of the wind speed ratio MAE and 2.10~2.90 of the wind direction MAE, which are about 34% and 44% lower than those of ABL IC, respectively.

(2) When the method of generating the initial conditions proposed in this study and the conventional initial conditions were mixed appropriately by wind direction sector, the overall CFD convergence time was confirmed to have been reduced by around 36% when employing 1 million computation cells and 16 wind direction sectors. Therefore, the proposed method is expected to make a substantial contribution to shortening the micro-siting project period for the design of a wind farm.

(3) The validity of the Reynolds number invariance of atmospheric wind flow over rough terrain was verified from the Hundhammerfjellet wind farm simulation with regard to the gradient wind speeds of 5 m/s, 10 m/s, and 15 m/s. The relative RMSE of the difference in wind speed between the Reynolds number $0.8 \times 10^8$ and $2.4 \times 10^8$ cases was only 0.65% and differences were found on the lee side of the steeply sloped, high terrain.

**Funding:** This work was conducted under the framework of the research and development program of the Korea Institute of Energy Research (B9-2414).

**Conflicts of Interest:** The authors declare no conflict of interest.

**Patent:** The main idea of this work has been registered to U.S. Patent 10,242,131 in March 2019.

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
