# Peer review of "A Method of Accelerating the Convergence of Computational Fluid Dynamics for Micro-Siting Wind Mapping"

_computation, doi:10.3390/computation7020022_

Round 1

Reviewer 1 Report

The manuscript entitled "Convergence acceleration method of computational fluid dynamics for micrositing wind mapping" deals with an interesting topic that I consider appropriate for the scientific objectives of the Computation journal.

Anyway, the manuscript in my opinion should be deeply revised in order to consider it acceptable for publication.

I have three main concerns:

1) The quality of the English is not adequate.

2) It is not acceptable that a journal manuscript has only 6 items in the references. The topic of the manuscript has been object of a considerable number of studies in the literature. Please expand considerably the references list and put your work in relation with the literature in a clear way. Sideways, this would enforce the statements about the novelty of this work with respect to the state of the art in the literature.

3) The initial condition selection criteria proposed by the author are based, if I have understood well, on the assumption that d\theta is small. However, when simulating 16 wind direction sectors: d\theta is not small. Did the author think about a sensitivity study? How would the precision of your results would change? 

Reviewer 2 Report

Please find attached my comments.

Round 2

Reviewer 1 Report

The author has addressed all my comments. I have no further remarks and therefore I recommend publication of this manuscript.

Reviewer 2 Report

my requirements have been fullfiled. Moderate English changes required before publication